# The Diagnostic Performance of Tumor Stage on MRI for Predicting Prostate Cancer-Positive Surgical Margins: A Systematic Review and Meta-Analysis

**DOI:** 10.3390/diagnostics13152497

**Published:** 2023-07-27

**Authors:** Yu Wang, Ying Wu, Meilin Zhu, Maoheng Tian, Li Liu, Longlin Yin

**Affiliations:** 1Department of Radiology, Sichuan Provincial People’s Hospital, University of Electronic Science and Technology of China, Chengdu 611731, China; wyu202307@163.com (Y.W.); liliu97@126.com (L.L.); 2Institute of Radiation Medicine, Sichuan Provincial People’s Hospital, University of Electronic Science and Technology of China, Chengdu 611731, China; 3Department of Radiology, Affiliated Hospital of North Sichuan Medical College, Nanchong 637000, China; yingwu199708@163.com; 4Department of Radiology, Huashan Hospital, Fudan University, Shanghai 200032, China; zhumeilin36@163.com; 5Department of Radiology, Affiliated Hospital of Southwest Medical University, Luzhou 646000, China; tmaoheng@163.com

**Keywords:** prostate cancer, magnetic resonance imaging, positive surgical margin, meta-analysis, systematic review

## Abstract

Purpose: Surgical margin status in radical prostatectomy (RP) specimens is an established predictive indicator for determining biochemical prostate cancer recurrence and disease progression. Predicting positive surgical margins (PSMs) is of utmost importance. We sought to perform a meta-analysis evaluating the diagnostic utility of a high clinical tumor stage (≥3) on magnetic resonance imaging (MRI) for predicting PSMs. Method: A systematic search of the PubMed, Embase databases, and Cochrane Library was performed, covering the interval from 1 January 2000 to 31 December 2022, to identify relevant studies. The Quality Assessment of Diagnostic Accuracy Studies 2 method was used to evaluate the studies’ quality. A hierarchical summary receiver operating characteristic plot was created depicting sensitivity and specificity data. Analyses of subgroups and meta-regression were used to investigate heterogeneity. Results: This meta-analysis comprised 13 studies with 3924 individuals in total. The pooled sensitivity and specificity values were 0.40 (95% CI, 0.32–0.49) and 0.75 (95% CI, 0.69–0.80), respectively, with an area under the receiver operating characteristic curve of 0.63 (95% CI, 0.59–0.67). The Higgins I2 statistics indicated moderate heterogeneity in sensitivity (I2 = 75.59%) and substantial heterogeneity in specificity (I2 = 86.77%). Area, prevalence of high Gleason scores (≥7), laparoscopic or robot-assisted techniques, field strength, functional technology, endorectal coil usage, and number of radiologists were significant factors responsible for heterogeneity (*p* ≤ 0.01). Conclusions: T stage on MRI has moderate diagnostic accuracy for predicting PSMs. When determining the treatment modality, clinicians should consider the factors contributing to heterogeneity for this purpose.

## 1. Introduction

According to the latest data for 2021, prostate cancer (PCa) is the most prevalent cancer among men, making it the second leading cause of death [1]. Although radical prostatectomy (RP) is the recommended treatment for prostate cancer, 20% of individuals who have RP surgery experience positive surgical margins (PSMs) that are detected by pathology [2]. This is a recognized negative prognostic indicator for PCa.

PSMs are generally defined as tumor cells reaching the inked surgical margin of the prostatectomy specimen [3]. PSMs are associated with local disease recurrence and distant metastasis, which may necessitate secondary treatment [4,5,6,7]. Therefore, a preoperative test for predicting clinically meaningful PSMs would be important for optimizing patients who require intraoperative frozen analysis or adjuvant treatment for obtaining better tumor control and avoiding unfavorable functional outcomes.

The digital rectal examination (DRE) is a standard procedure for diagnosing and staging PCa. However, the importance of precise DRE staging remains debatable. In recent years, multiparameter magnetic resonance imaging (mpMRI) has been clinically used in the diagnosis of PCa, including in tumor detection, localization, localized staging, and determining tumor aggressiveness. MRI is regarded as the best available imaging tool for assessing tumor stage (T stage) in clinical practice [8,9]. High clinical T stage disease (T3 stage), including extraprostatic extension—namely, extracapsular extension, and seminal vesicle infiltration—determined by MRI has been significantly associated with more prevalent PSMs [10] and frequently linked to a higher risk of biochemical recurrence as well as metastatic disease [11], which might be crucial for planning RP approaches and nerve-sparing surgery.

For this reason, preoperative MRI not only focuses on predicting the T stage, but it is also important for tailoring surgical concepts for patients and guiding surgeons to achieve negative surgical margins [12], which is significant for survival, prognosis, and recurrence. Relevant scoring systems have been developed to predict PSMs [13,14], and the T3 stage on MRI was considered an independent predictor of PSM after RP [14]. However, the association between the T stage on MRI and PSM remains controversial. Furthermore, there has been no systematic evaluation of the diagnostic significance of MRI for predicting PSMs.

Therefore, this meta-analysis aimed to assess the diagnostic performance of the T3 stage as a predictive indicator of PSMs in patients with PCa.

## 2. Materials and Methods

This meta-analysis was conducted and written according to PRISMA (Preferred Reporting Items for Systematic Reviews and Meta-Analyses) guidelines. The study protocol was registered with INPLASY (INPLASY202370012).

### 2.1. Literature Search

Two researchers employed a systematic search strategy using PubMed, the Cochrane Library, and Embase to find articles reporting on studies evaluating the value of MRI to predict PSMs. Articles published between 1 January 2000 and 31 December 2022 were included in the searches. We applied the search strategy based on medical topic headings, free words, and their variants. The literature retrieval process had no language restrictions. The detailed retrieval strategy is described in the Appendix A. The reference lists for articles and reviews containing combinations of search strings were checked.

### 2.2. Inclusion Criteria

We included articles reporting on diagnostic accuracy studies if (1) accuracy was assessed for PSMs using non-organ localized diseases observed on MRI as the index test among PCa patients, (2) if the histopathology of RP specimens served as the reference standard, (3) if studies had enough information to develop a 2 × 2 table to evaluate the diagnostic accuracy, and (4) the if article type was an “original article” or equivalent.

### 2.3. Exclusion Criteria

The exclusion criteria were (1) studies with a variety of topics, such as the diagnostic usefulness of other MRI findings for PSM prediction, (2) the reference standard was not RP specimen, (3) insufficient necessary data for meta-analytic pooling, and (4) publication type other than the original article.

### 2.4. Data Extraction

The extracted data include the first author, year of publication, study characteristics, demographic characteristics, imaging characteristics, numbers of true/false positives, and true/false negatives.

### 2.5. Methodologic Quality Assessment

Study quality was evaluated utilizing the QUADAS-2 (Quality Assessment of Diagnostic Accuracy Studies 2) tool [15]. The risk of bias for each study was assessed in four ways: (1) patient selection, (2) index test, (3) reference standard, and (4) flow and timing. Bias risk was rated as low, high, or unclear.

### 2.6. Data Synthesis and Analysis

Diagnostic tables including true positives, false negatives, false positives, and true negatives were used to calculate sensitivity and specificity. The bivariate random effects model was used to assess the diagnostic efficacy indices including pooled sensitivity, specificity, and their 95% confidence intervals (CIs) [16]. The hierarchical receiver operating characteristic (HSROC) summary curve analysis with the area under the receiver operating characteristic curve (AUC) was plotted to exhibit the diagnostic precision [17]. Deeks’ funnel plot asymmetry test was used to identify publication bias [18]. Significant heterogeneity was indicated by the following criteria: *p* < 0.05 in Cochrane’s Q test and an I2 ratio >50%. For evaluation of the heterogeneity between studies, the potential effects of several covariates were investigated by subgroup analysis. The covariates included (1) study design (retrospective vs. prospective), (2) area (Asia vs. non-Asia), (3) use of minimally invasive techniques (laparoscopic vs. non-laparoscopic), and robot assistance (robot-assisted vs. not robot-assisted), (4) prevalence of high Gleason scores (≥7) on the biopsy (≥50% vs. <50%), (5) magnetic field strength (3 T vs. not 3 T), (6) use of endorectal coils (ERCs), (7) functional MRI technology (MRI sequences using apparent diffusion coefficients and dynamic contrast enhancement (DCE) vs. not), (8) the number of radiologists (multiple vs. single), and (9) the number of cases (≥150 vs. <150).

One reviewer (W.Y.) conducted all analyses using Stata 17.0 (StataCorp, College Station, TX, USA), with *p* < 0.05 denoting statistical significance.

## 3. Results

### 3.1. Literature Search and Article Selection

The PRISMA flow diagram depicts the article screening process, as shown in Figure 1.

Initially, after systematic searching, 544 articles were obtained in the PubMed, Embase, and Cochrane Library databases. Fifty-seven duplicates were removed. The remaining 487 titles and abstracts were filtered, leaving 54 potentially eligible papers for full-text screening. After the full-text review, 13 articles [4,10,12,19,20,21,22,23,24,25,26,27,28] were acceptable for this systematic review. The reasons for exclusion were (1) insufficient data to generate 2 × 2 tables (*n* = 2), (2) evaluation of other MRI findings as predictors of PSMs (*n* = 25), and (3) the reported study did not assess the outcomes of interest (*n* = 14). Ultimately, 13 articles (3924 patients) were included in this meta-analysis.

### 3.2. Study Characteristics

Table 1 summarizes the demographic characteristics, first author, sample size, year of publication, area, age, prostate-specific antigen levels, number of patients with high biopsy Gleason score and proportion and mode of biopsies, and DRE results. Study characteristics included study design, consecutive patient selection, NeuroSAFE techniques, whether minimally invasive techniques were used, reference standard, the interval between imaging and surgery, and surgeons (Table 2). Table 3 summarizes imaging characteristics including magnetic field strength, number and experience of radiologists, blinding, use of ERCs, MRI vendors, and imaging sequence details, including functional DCE MRI and diffusion-weighted imaging (DWI), as well as T2-weighted imaging (T2WI).

The sample sizes of individual studies ranged from 48 to 1145. One study was prospective in design [25]; the remaining twelve were retrospective. ERCs were used in three studies [19,24,28]. For blinding, two papers reported that the radiologists were aware that patients had biopsy evidence of PCa [19,24], and three articles did not specify blinding [4,12,20]. In four studies, only 1.5 T scanners were used [12,23,24,28], compared with six studies wherein MRI was performed using 3.0 T scanners. One study used both 1.5 T and 3.0 T scanners [10]. The use of the NeuroSAFE technique was mentioned in four studies [12,20,21,24].

### 3.3. Quality Assessment

In terms of study quality, the overall quality was moderate. The majority of research in the field of patient selection had retrospective designs, which were seen as having an unclear risk of bias. Due to the reader’s exposure to the reference tests during the interpretation of the index tests in two trials, the index domain was of high concern [19,24]. Six studies were labeled as unclear regarding the applicability of index tests due to a lack of information presented about index test characteristics, interpretation, and blinding [4,10,12,20,22,28]. All studies were deemed to have a low risk of bias in the reference test domain since they all used RP specimens as the reference standard. For the flow and timing domain, all of the included studies were found to have a high risk of bias. The quality of the included investigations determined by QUADAS-2 rules is shown in Figure 2.

### 3.4. Diagnostic Performance of Non-Organ-Confined Disease on MRI for the Detection of PSMs

According to forest plots (Figure 3), the pooled sensitivity and specificity were 0.40 (95% CI, 0.32–0.49) and 0.75 (95% CI, 0.69–0.80), respectively. Figure 4 shows HSROC plots with an AUC of 0.63 (95% CI, 0.59–0.67). Heterogeneity was found, according to the Q test (*p* < 0.05). The Higgins I2 statistics revealed significant heterogeneity for specificity (I2 = 86.77%) and moderate heterogeneity for sensitivity (I2 = 75.59%).

### 3.5. Subgroup Analysis and Meta-Regression

Table 4 illustrates the results of the investigation of the causes of the pooled variability using subgroup analysis. In general, sensitivity was comparable (I2 = 75.59%), but specificity (I2 = 86.77%) showed significant variations.

Among all the investigated covariates, eight covariates (all but study design and area) were revealed as significant factors contributing to heterogeneity (*p* < 0.01).

For 12 retrospective studies, the pooled sensitivity and specificity were 0.40 (95% CI, 0.31–0.48) and 0.79 (95% CI, 0.63–0.95), respectively. In eight studies where the races of the participants were non-Asian, the sensitivity and specificity were 0.41 (95% CI, 0.30–0.52) and 0.74 (95% CI, 0.67–0.80), respectively. The pooled sensitivity and specificity were 0.48 (95% CI, 0.38–0.57) and 0.65 (95% CI, 0.58–0.73) when the prevalence of high biopsy Gleason scores was >50%. In nine studies that investigated laparoscopic procedures, the pooled sensitivity and specificity were 0.43 (95% CI, 0.33–0.52) and 0.72 (95% CI, 0.65–0.78), respectively. In six studies investigating robot-assisted procedures, the pooled sensitivity and specificity were 0.44 (95% CI, 0.33–0.56) and 0.72 (95% CI, 0.64–0.81). In seven studies using 3.0 T as the field strength, the pooled sensitivity and specificity were 0.42 (95% CI, 0.29–0.54) and 0.75 (95% CI, 0.68–0.82), respectively. In seven studies that used functional sequences, the pooled sensitivity and specificity were 0.47 (95% CI, 0.36–0.58) and 0.71 (95% CI, 0.65–0.77), respectively. Three studies used ERCs, and among these, the pooled sensitivity and specificity were 0.24 (95% CI, 0.10–0.39) and 0.81 (95% CI, 0.73–0.89), respectively.

### 3.6. Publication Bias Assessment

Figure 5 shows a *p*-value of 0.26 for the slope coefficient, indicating a low probability of publication bias.

## 4. Discussion

We performed a meta-analysis focusing on the diagnostic value of the high clinical T stage on MRI as a predictive indicator for PSMs. As far as we are aware, this is the first assessment of the diagnostic precision of preoperative MRI staging to predict PSMs. There were 13 articles included with a total of 3924 participants. Our meta-analysis demonstrated the T3 stage on MRI to have a moderate diagnostic performance for predicting pathologically positive margins with suboptimal sensitivity but relatively high specificity. The pooled sensitivity and specificity were 0.40 (95% CI, 0.32–0.49) and 0.75 (95% CI, 0.69–0.80), respectively, with an AUC of 0.63 (95% CI, 0.59–0.67). Overall, MRI had a relatively low diagnostic performance for the presence of PSMs on specimens after RP. To some extent, this reflects the resolution limitations of multiparametric MRI pulse sequences and our practice of conservative readings [29]. The T3 stage is a significant pathological characteristic of PCa, which raises the likelihood of PSMs and biochemical recurrence [30]. However, identifying the T3 stage from MRI is often considered to be subjective. The criteria depend on the reader’s expertise, especially the extracapsular extension score—which is mainly based on qualitative and subjective features—so relatively few reproducible imaging criteria are involved [12,31]. A weak interobserver agreement may also contribute to suboptimal diagnostic performance.

Notably, the T3 stage on preoperative MRI is not linearly related to the presence of histopathological PSMs. However, the T3 stage might be important because interpreting these findings for urologists may alter the anatomical plane and surgical approaches. Knowledge of the location and extent of a tumor on MRI may enable a surgeon to more precisely decide the treatment modality. When nerve sparing is the goal, rates of PSMs tend to be higher, so surgeons prefer to perform non-nerve-preserving techniques to achieve negative surgical margins when non-organ-localized diseases are observed on MRI [10,19,24]. Therefore, the benefit of MRI staging lies in the correct selection of patients for neurovascular bundle (NVB)-preserving surgery, rather than technically preventing PSMs [23]. When selecting the best patients for active surveillance or selecting RP candidates, high sensitivity is required to preserve NVBs. On the other hand, taking a conservative approach could result in PCa patients being unnecessarily excluded from NVB-sparing surgery; the relatively high specificity determined by our analysis may provide some auxiliary information for clinicians.

Our meta-analysis demonstrated that prospective experimental designs appear to be more sensitive and specific than retrospective designs. The use of high field strength (3.0 T) appears to be useful for sensitivity improvement. Unsurprisingly, high field strengths may theoretically reflect extracapsular tumors better than low field strengths due to the detection of subtle capsular irregularities or small extraprostatic tumors that demand a high spatial resolution for detection, and this is more available at higher magnetic strengths, despite the claim by May et al. (38) that using ERC can significantly increase the signal-to-noise ratio and may lead to improved discrimination of surgical margin status [26]. Some preliminary studies have shown that MRI with ERC MRI cannot accurately forecast PSM, it is only effective for a small subset of high-risk high-vascular tumors, for which radiological interpretations vary widely [32,33,34]. Similarly, our analysis indicated that ERCs do not appear to be beneficial for increased sensitivity or specificity compared with surface magnets. Among the studies included in our analysis, six of the seven that used field strengths of 3.0 T did not use ERCs. Notably, 3.0 T phased-array MRI has been shown to be equivalent to 1.5 T intrarectal MRI with no significant loss of image quality [35]. Meanwhile, long patient preparation means significant time-to-scan protocols [36]. These factors undermine the utility of ERCs, leading many centers to no longer routinely use ERCs for diagnostic protocols.

We analyzed the influence of MRI-specific functional sequences. When additional functional technology (including DWI and DCE) was used, sensitivity was significantly improved compared with when only T2WI was used (0.47 vs. 0.29). Possible causes might be that mpMRI provides additional details regarding gland anatomy and tumor location and allows for a thorough assessment of anatomical obstacles and unfavorable tumor features that raise the likelihood of PSM [37]. By using a combination of T2W-MRI with functional sequences, the delineation of the tumor and normal prostate boundaries can be improved [31].

Laparoscopic and robot-assisted minimally invasive techniques have grown incredibly popular during the past 10 years. With robot-assisted surgery, the use of high-resolution cameras with three-dimensional imaging and robotic arms, which enable surgeons to execute more precise dissection of anatomic structures, may improve the preservation of functional structures and reduce PSMs [38]. However, when minimally invasive techniques are used, there is a lack of tactile examination of the prostate to assess the degree of aggressiveness of the malignancy, and excision may result in PSMs and lead to an increased reliance on imaging to determine staging. In the absence of tactile feedback, MRI becomes an effective technique for assessing the degree of nerve retention [39]. In the present analysis, minimally invasive techniques were observed to cause significant differences in specificity and sensitivity, which implied positive margins in laparoscopic RP, or robot-assisted RP may differ from the negative prognostic significance of positive margins associated with open RP.

The prevalence of high-risk tumors is another important factor in the causation of heterogeneity. Studies with more high-Gleason tumors (>50%) had significantly higher sensitivity and specificity values compared with those without (<50%). This could be because high-grade PCa was associated with lower prostate volume and longer diameters of lesions on MRI [40,41]. PCa is detected earlier in patients with enlarged prostates because of the elevated levels of prostate-specific antigen produced by the enlarged tissue, and PCa diagnosed in small glands may be more aggressive and associated with more unfavorable histopathological findings. Patients with small prostates had a higher rate of PSMs [42]. More chaotic gland lumen typically occurs in tissues with higher Gleason scores [43]. This means that the higher the Gleason score, the more disrupted the glandular structure will be, resulting in a more disorganized glandular lumen, which can be reflected by ADC and DWI [44].

A recent meta-analysis [45] noted that significantly lower PSM rates were reported after 50–60 cases, reaching a plateau at 150–350 cases. Compared to low-volume centers (cases <150), high-volume centers (cases ≥150) appear to have improved diagnostic sensitivity and specificity. This may prove that high-volume compared to low-volume centers are generally associated with more favorable surgical outcomes.

Our analysis included some drawbacks. First, many studies did not provide enough details about all study characteristics, and we could not adequately account for the heterogeneity. For example, information on blind and image interpretation methods is often not available when interpreting index tests. Second, subjectivity is unavoidable to some extent when interpreting the T3 phase on MRI because it is observer-dependent. Features that are more objective, quantitative, and repeatable might be more appropriate.

## 5. Conclusions

Despite the poor sensitivity of the T stage on preoperative MRI for predicting pathological PSMs, its relatively high specificity may have additional value in helping to select candidates for active surveillance or function-preserving therapy by avoiding underestimating the disease. Despite the reliability of MRI in identifying cancer lesions and boundaries, determining the aggressiveness of prostate cancer and predicting the outcome of the surgery remains challenging. Therefore, efforts should focus on further refining the characteristics of PSMs to help identify patients who are most prone to disease development and progression. Further studies are warranted to investigate more objective and reproducible variables as potential indicators of surgical margin status.

## Figures and Tables

**Figure 1 diagnostics-13-02497-f001:**
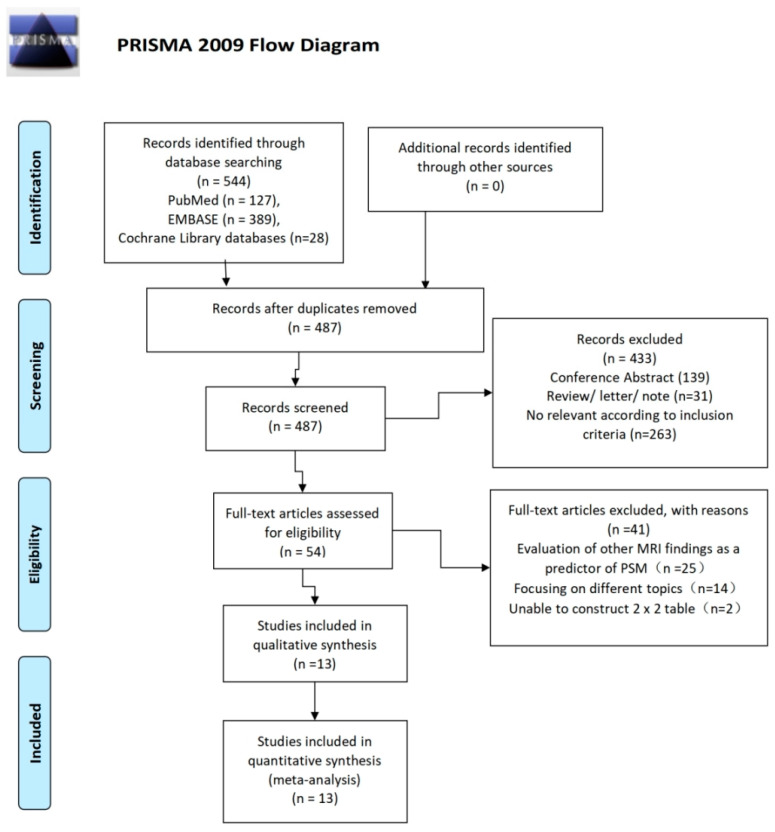
Flow diagram describing the study selection process for the meta-analysis.

**Figure 2 diagnostics-13-02497-f002:**
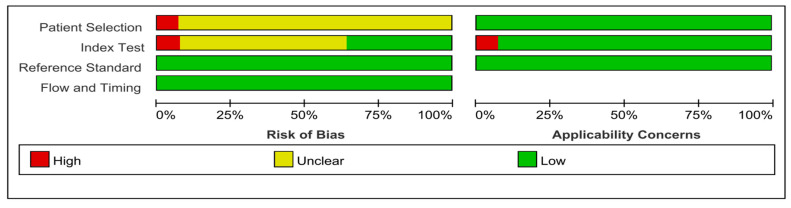
Study bias assessment.

**Figure 3 diagnostics-13-02497-f003:**
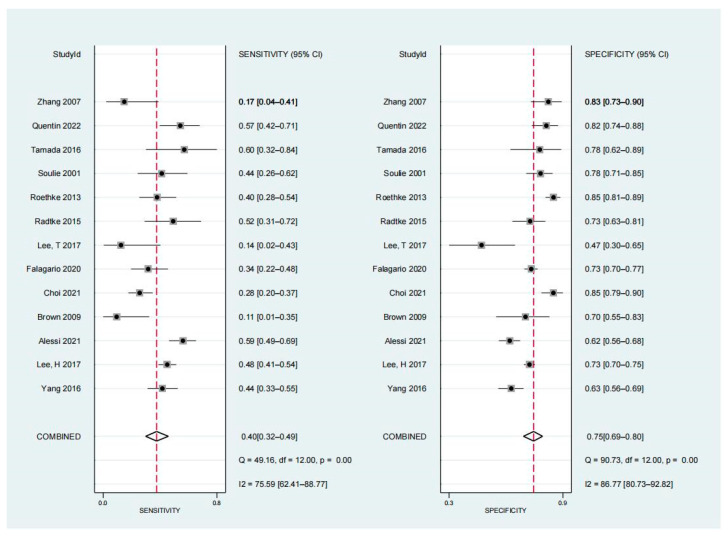
Coupled forest plots summary of sensitivity and specificity [4,10,12,19,20,21,22,23,24,25,26,27,28]. Numbers are pooled estimates with 95% CI in parentheses. Corresponding heterogeneity statistics are provided at bottom right corners. Horizontal lines indicate 95% CIs. CIs, confidence intervals.

**Figure 4 diagnostics-13-02497-f004:**
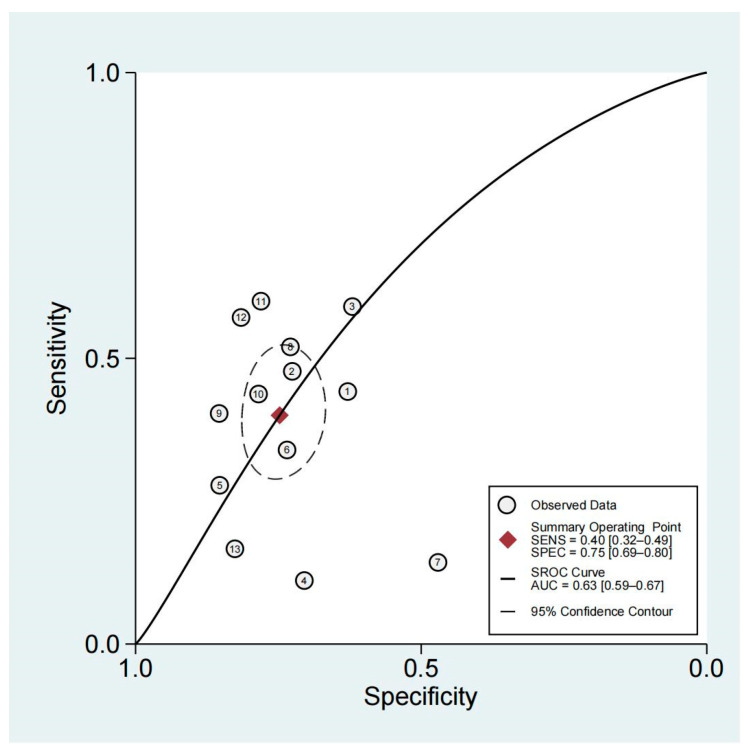
HSROC curve of the diagnostic performance of MRI for PSM prediction in prostate cancer patients.

**Figure 5 diagnostics-13-02497-f005:**
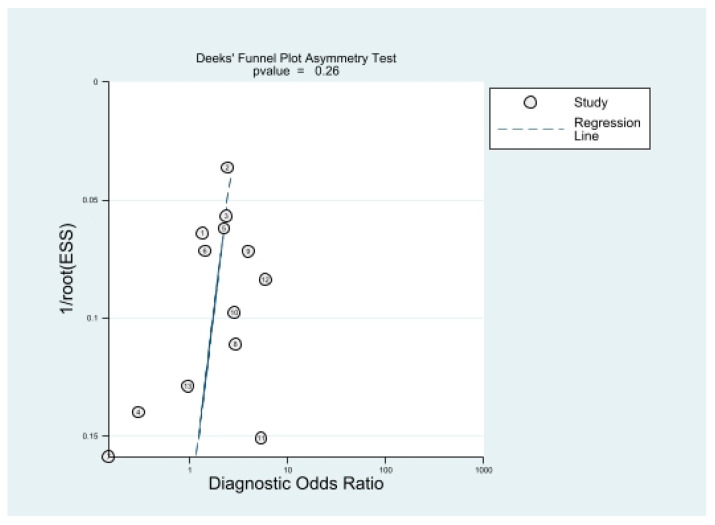
Deeks’ funnel plot. The likelihood of publication bias was low with a *p*-value of 0.26 for the slope coefficient.

**Table 1 diagnostics-13-02497-t001:** Demographic characteristics of the included studies.

Author	Year	Country	*N*	Age (Years), Median (Range)	PSA (ng/mL), Median (Range)	No. of Biopsies with Gleason Score ≥7	Proportion and Mode of Biopsies	DRE (+)
Soulie [25]	2001	France	176	64.2 * (49–74)	10.9 * (1.2–39)	NA	97.2%, TRUS	NA
Zhang [28]	2007	China	110	62.2 * (NA)	8.1 * (NA)	NA	All, NA	NA
Alessi [12]	2021	Italy	400	65 (NA)	6.3 (NA)	220	All, NA	NA
Lee, H. [10]	2017	South Korea	1145	65 (NA)	6.1 (NA)	594	All, NA	NA
Radtke [23]	2015	Germany	132	66 (NA)	8.2 (NA)	NA	All, MRI/TRUS-fusion	28
Yang [4]	2016	China	296	70 (51–80)	12.6 (2.9–180)	196	ALL, TRUS	NA
Lee, T. [22]	2017	Canada	48	60.9 * (NA)	9.7 * (NA)	36	ALL, TRUS	NA
Falagario [21]	2020	Italy	664	63 (NA)	6 (NA)	NA	All, NA	NA
Tamada [26]	2016	Japan	56	69 (54–84)	7.26 (4.09–24.5)	49	All, NA	15
Roethke [24]	2013	Germany	385	62.7 * (42–77)	8.9 * (0.4–52.5)	NA	All, NA	NA
Brown [19]	2009	United States	62	58 * (42–73)	9.3 * (0.5–30.5)	NA	All, NA	NA
Quentin [27]	2022	Germany	179	NA	NA	NA	All, NA	NA
Choi [20]	2021	South Korea	271	64.9 * (NA)	7.9 * (NA)	78	All, NA	NA

* = mean. Abbreviations: PSA = prostate-specific antigen; NA = not available; TRUS: transrectal ultrasonography; DRE (+): digital rectal examination positive.

**Table 2 diagnostics-13-02497-t002:** Study characteristics of the included studies.

First Author	Study Design	Patient Enrollment	Index	NeuroSAFE Technique	Radical Prostatectomy Technique	Interval between Biopsy and MRI	No. of Surgeons	Reference
Soulie [25]	Prospective	Consecutive	T3	NA	RRP	2 to 9 w	2	RP
Zhang [28]	Retrospective	Consecutive	T3	NA	NA	3 to 10 w	1	RP
Alessi [12]	Retrospective	Consecutive	ECE score ≥4	yes	RARP	NA	Multiple	RP
Lee, H. [10]	Retrospective	Consecutive	T3	NA	RARP, LRP, ORP	NA	NA	RP
Radtke [23]	Retrospective	Consecutive	ECE score ≥4	NA	RARP, RRP	NA	4	RP
Yang [4]	Retrospective	Consecutive	EPE	NA	LRP	NA	1	RP
Lee, T. [22]	Retrospective	Consecutive	ECE	NA	LRP, RARP, ORP	At least 6 w	1	RP
Falagario [21]	Retrospective	Consecutive	EPE	Yes	NA	4 w	Multiple	RP
Tamada [26]	Retrospective	Consecutive	ECE	NA	LRP, RRP	NA	NA	RP
Roethke [24]	Retrospective	Consecutive	ECE	Yes	RRP	At least 6 w	1	RP
Brown [19]	Retrospective	Consecutive	T3	NA	LRP, ORP	At least 8–10 w	2	RP
Quentin [27]	Retrospective	Consecutive	EPE	NA	RARP	NA	4	RP
Choi [20]	Retrospective	Consecutive	ECE	Yes	RALP	NA	1	RP

Abbreviations: RP = radical prostatectomy; RRP = retropubic radical prostatectomy; LRP = laparoscopic radical prostatectomy; RARP = robot-assisted radical prostatectomy; ORP = open retropubic radical prostatectomy; NA = not available; EPE = extraprostatic extension; ECE = extracapsular extension.

**Table 3 diagnostics-13-02497-t003:** Magnetic resonance imaging characteristics of the included studies.

Author	Field Strength (T)	Blinded	No. of Radiologist	Experience of Radiologist (Years/Numbers)	Endorectal coil	MR Techniques	Vendor
Soulie [25]	1.0	Yes	Single	NA	No	T1, T2	Siemens
Zhang [28]	1.5	Yes	Single	NA	Yes	T1, T2	GE
Alessi [12]	1.5	NA	Three	10 y, 8 y, 6 y/>500	No	T2, DWI, DCE	Siemens
Lee, H. [10]	1.5–3.0	Yes	Two	NA	No	T2, DWI, ADC	Philips
Radtke [23]	3.0	Yes	Multiple	7 y, >10 y	No	T2, DWI, DCE	Siemens
Yang [4]	NA	NA	NA	NA	NA	NA	NA
Lee, T. [22]	3.0	Yes	Single	>10 y	No	T2, DWI, DCE	Philips
Falagario [21]	3.0	NA	Single	NA	No	T2, DWI, DCE	Siemens
Tamada [26]	3.0	Yes	Two	7 y, 16 y	No	T1, T2, DWI, DCE	Toshiba
Roethke [24]	1.5	Yes	Two	4–14 y/300–1000	Yes	T2	Siemens
Brown [19]	1.5	No	Multiple	1–20 y	Yes	T1, T2	GE
Quentin [27]	3.0	Yes	Two	>10 y	No	T1, T2, ADC, DCE	Siemens
Choi [20]	3.0	NA	NA	NA	NA	NA	NA

DWI = diffusion-weighted (magnetic resonance) imaging; DCE = dynamic contrast-enhanced (magnetic resonance imaging); ADC = apparent diffusion coefficient; NA = not available.

**Table 4 diagnostics-13-02497-t004:** Meta-regression analyses stratified by multiple variables.

Variable	No. of Studies	Category	Sensitivity		Specificity		LRT Chi-Square	P (Joint Model)
			Pooled Value (95% CI)	*p*	Pooled Value (95% CI)	*p*		
Study design	1	Prospective	0.44 (0.14–0.74)	0.7	0.79 (0.63–0.95)	0.42	0.03	0.86
	12	Retrospective	0.40 (0.31–0.48)		0.74 (0.69–0.80)			
Area	8	Non-Asian	0.41 (0.30–0.52)	0.34	0.74 (0.67–0.80)	<0.01	0.40	0.82
	5	Asian	0.39 (0.26–0.51)		0.77 (0.69–0.84)			
Prevalence of biopsy Gleason score (≥7)	5	>50%	0.48 (0.38–0.57)	0.02	0.65 (0.58–0.73)	<0.001	106.04	<0.01
	1	<50%	0.28 (0.13–0.42)		0.85 (0.76–0.94)			
Robotic-assisted	6	Yes	0.44 (0.33–0.56)	0.36	0.72 (0.64–0.81)	0.01	29.21	<0.01
	5	No	0.40 (0.27–0.53)		0.76 (0.68–0.85)			
Laparoscopic	9	Yes	0.43 (0.33–0.52)	0.67	0.72 (0.65–0.78)	<0.001	31.24	<0.01
	2	No	0.42 (0.22–0.62)		0.82 (0.74–0.91)			
Field strength	7	3.0 T	0.42 (0.29–0.54)	0.28	0.75 (0.68–0.82)	<0.01	17.07	<0.01
	5	1.5 T or 1 T	0.36 (0.22–0.50)		0.77 (0.69–0.85)			
Functional technology	7	Yes	0.47 (0.36–0.58)	0.04	0.71 (0.65–0.77)	<0.001	42.2	<0.01
	4	No	0.29 (0.15–0.43)		0.80 (0.74–0.87)			
Endorectal coil	3	Yes	0.24 (0.10–0.39)	0.38	0.81 (0.73–0.89)	0.08	43.09	<0.01
	8	No	0.47 (0.37–0.57)		0.72 (0.66–0.78)			
No. of radiologists	7	Multiple	0.48 (0.39–0.57)	0.02	0.75 (0.69–0.82)	0.02	37.3	<0.01
	4	Single	0.30 (0.18–0.42)		0.73 (0.64–0.82)			
No. of cases	8	≥150	0.44 (0.35–0.53)	0.04	0.76 (0.70–0.82)	0.03	3.36	0.19
	5	<50	0.31 (0.18–0.44)		0.72 (0.63–0.82)			

CI = confidence interval, LRT = likelihood ratio test.

## Data Availability

The original contributions presented in the study are included in the article/Appendix A. Further inquiries can be directed to the corresponding author.

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
