# Peer review of "The Diagnostic Performance of Tumor Stage on MRI for Predicting Prostate Cancer-Positive Surgical Margins: A Systematic Review and Meta-Analysis"

_diagnostics, 2023, doi:10.3390/diagnostics13152497_

Round 1

Reviewer 1 Report

MRI is nowadays used to select patients for biopsies and in this study patients had MRI post-biopsies which could influence the MRI reading. What was interval between biospies and MRI. How were biopsies taken and how many. Complications following biopsies?

What were results of digital rectal examination and was this perfumed in all patients?

It is generally accepted that MRI is not really helpful for staging.

What is important to know if R+ is found and if neurosafe procedures were performed.

Experience of MRI readers is important, how many MRI readings were performed

small changes

Author Response

Dear reviewer,

Thank you very much for your comments concerning our manuscript ID: diagnostics-2474808. Those comments are valuable and very helpful. We have read through comments carefully and have made corrections. Our point-by-point responses please see the attachment.

Reviewer 2 Report

The authors performed a meta-analysis of 13 studies to determine whether MRI performed before prostatectomy can predict positive surgical margins (PSM) in surgical specimens. Overall, MRI did not adequately predict PSM.

PSM is by no means absolute or unchangeable, and is influenced in part by surgical technique. In general, it can be assumed that PSM will decrease to a certain degree in institutions and surgeons with a large number of cases. Therefore, in the analysis of Table 4, it is necessary to add a category to categorize the number of cases in each study.

The term "laparoscopic" sometimes includes both pure laparoscopic and robotic, and sometimes only pure laparoscopic. It is desirable to avoid these confusions.

From Table 4, it reads as if 1.5T is more specific than 3T. This may be contrary to the discussion section (lines 237-239).

Author Response

Dear reviewer,

Thank you very much for your comments concerning our manuscript ID: diagnostics-2474808. Those comments are valuable and very helpful. We have read through comments carefully and have made corrections.

Please see the attachment for our modification and reply.

Round 2

Reviewer 1 Report

comments well addressed

NA